

# Fermionization of a few-body Bose system immersed into a Bose-Einstein condensate

Tim Keller⋆, Thomás Fogarty and Thomas Busch

Quantum Systems Unit, Okinawa Institute of Science and Technology Graduate University,
Onna-son, Okinawa 904-0495, Japan

⋆ tim.keller@oist.jp

## Abstract

We study the recently introduced self-pinning transition [Phys. Rev. Lett. 128, 053401 (2022)] in a quasi-one-dimensional two-component quantum gas in the case where the component immersed into the Bose-Einstein condensate has a finite intraspecies interaction strength. As a result of the matter-wave backaction, the fermionization in the limit of infinite intraspecies repulsion occurs via a first-order phase transition to the self-pinned state, which is in contrast to the asymptotic behavior in static trapping potentials. The system also exhibits an additional superfluid state for the immersed component if the interspecies interaction is able to overcome the intraspecies repulsion. We approximate the superfluid state in an analytical model and derive an expression for the phase transition line that coincides with well-known phase separation criteria in binary Bose systems. The full phase diagram of the system is mapped out numerically for the case of two and three atoms in the immersed component.

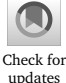

# 1 Introduction

Quantum phase transitions are one of the most interesting aspects of quantum many-body physics [1] and they underlie many of the novel states and unique properties that allow neutral atoms in optical lattice potentials to be used as quantum simulators of condensed matter physics [2–5]. A paradigmatic example of this is the superfluid to gapped Mott insulator or pinned-state transition that happens in a one-dimensional quantum gas that is subject to an external lattice potential of arbitrary small lattice strengths beyond a critical value of the interparticle repulsion [6, 7]. A mapping between the gas parameters and the Luttinger liquid parameter [8] allows finding this critical value from the renormalization group theory treatment of the sine-Gordon model [9]. We have recently shown how the same physics can be observed without the need for a commensurate lattice potential, but by immersing the strongly-correlated quantum gas into a weakly-correlated background such as a Bose-Einstein condensate (BEC) instead [10]. The immersed component is able to create its own commensurate matter-wave trapping potential via the backaction with the background gas, leading to an equivalent quantum phase transition.

In both cases the transition is of second order and therefore continuous. First-order or discontinuous transitions are a lot less common in cold atomic systems. Such transitions are characterized by metastable states in which a system can remain even after crossing the transition [11] and there is a growing interest in the form of proposals for cold atom quantum simulators of the early universe [12, 13], which might be able to investigate the 'fate of the false vacuum' [14], i.e. the decay from a metastable state to the true ground state. Metastable states also give rise to hysteresis and the effects associated with discontinuous phase transitions have already been observed in cold atom experiments, for example for an ultracold atomic gas in a double-well potential [15], in spinor BECs [16], in a driven one-dimensional (1D) optical lattice [17] and also in imbalanced quasi-1D Bose-Bose mixtures [18].

One-dimensional Bose-Bose mixtures have seen considerable interest, in particular over the past twenty years [19–21]. If one interprets the two components as representations of the spin-up and spin-down state in a pseudo-spin 1/2 system, it has been shown that they can support spin-waves [22, 23] and also develop a spontaneous population imbalance, an equivalent of ferromagnetism [24, 25], which is suppressed at finite temperatures [26]. Eisenberg and Lieb showed that in general the ground state of such interacting bosons with spin is always fully polarized, i.e. ferromagnetic [27]. Furthermore, they can be used to demonstrate a hallmark effect in 1D electronic systems, which is the separation of single-particle spin and charge excitations into two distinct collective branches as a result of the dimensionality [28].

A Luttinger liquid approach predicts collapse and pairing phenomena for attractive inter-component interactions similar to 1D Bose-Fermi mixtures, depending on the density regime and intracomponent interactions [29, 30], while for repulsive intercomponent interactions a miscibility criterion identical to the well-known 3D case determines the regime of phase separation [20]. Furthermore, exact solutions have been found for a 1D Bose-Bose mixture on a ring with equal densities and identical intraspecies and interspecies interaction strengths [31] and similar to Bose-Fermi mixtures, a supersolid state was also predicted for weak interspecies but nearly hard-core intraspecies repulsion in a balanced, harmonically trapped Bose mixture with densities incommensurate with the additional lattice potential [32].

In the few-atom regime, a balanced repulsive binary Bose mixture is characterized by six different limiting cases of either vanishing ('BEC limit') or infinite ('TG limit') interactions, quantified by $g_{ij}$ [33]. Along with the phase separated regime there is also 'composite fermionization' [34] ($g_{ii} \to 0$, $g_{ij} \to \infty$) and 'full fermionization' ($g_{ii}, g_{ij} \to \infty$), i.e. a single-component Fermi gas equivalent. Finally, more recent developments include the extension of the famous droplets in 3D Bose-Bose mixtures, stabilized purely by quantum fluctuations,

to the one-dimensional case at zero temperature [35, 36], in the presence of spin-orbit coupling [37], and at finite temperature [38]. There, the self-bound droplets are also predicted to have a bright solitonic shape, i.e. a density $n(x) \sim \cosh^{-2}(x)$, similar to the states found in Ref. [10]. Particularly, for imbalanced mixtures like the one studied in this work, similar structures in the form of composite 'bright-gray' solitons [39] or double domain-wall solitons [40] have been predicted.

In this work we show that an imbalanced one-dimensional Bose-Bose mixture consisting of a few-body bosonic system immersed into a much larger Bose-Einstein condensed background also gives rise to an apparent first-order phase transition between a coherent superfluid state and the insulating self-pinned state studied in Ref. [10] as the intraspecies repulsion is increased beyond a critical value going towards the Tonks-Girardeau limit. The paper is organized as follows. In Section 2 we introduce the system and the coupled Schrödinger equations we use to describe the two components. In Section 3 we numerically study the fermionization process of the immersed component as a function of the intraspecies interaction mainly in terms of the energy and density of the system. We use the analytical model presented in Ref. [10] as benchmark for the limiting cases of vanishing ($g \to 0$) and infinite ($g \to \infty$) intraspecies repulsion and expand the model to the superfluid state which persists for finite values between $0 < g < \infty$. Here we also derive an analytical approximation for the phase transition line and compare to numerical results. In Section 4 we numerically calculate the phase diagrams in terms of the coherence for two and three atoms in the immersed component before concluding in Section 5.

## 2 Model

We consider a strongly imbalanced two-component quantum gas in a quasi-one-dimensional setting. The majority component is a Bose-Einstein condensate (BEC) of $N_c$ particles, described in the mean-field limit by a macroscopic wave function $\psi(x)$. The minority component immersed into the BEC consists of $N \ll N_c$ particles described by a full many-particle wave function $\Phi(\mathbf{x} = x_1, x_2, \ldots, x_N)$. The system is studied at zero temperature and therefore we can describe all interactions by point-like pseudo-potentials that only depend on the interspecies and intraspecies scattering lengths. We model the interactions between the two components by a simple density coupling, which is valid when the interspecies interaction is assumed to be weak. This leads to the coupled Schrödinger equations

$$i\hbar\dot{\psi}(x) = \left[ -\frac{\hbar^2}{2m}\frac{\partial^2}{\partial x^2} + g_m|\Phi|^2 + g_c|\psi|^2 \right]\psi(x), \tag{1a}$$

$$i\hbar\dot{\Phi}(\mathbf{x}) = \left[ \sum_{l=1}^{N} -\frac{\hbar^2}{2m}\frac{\partial^2}{\partial x_l^2} + g_m|\psi|^2 + g\sum_{k<l}^{N}\delta(|x_k - x_l|) \right]\Phi(\mathbf{x}). \tag{1b}$$

The intraspecies interaction strengths in the minority and majority component are labeled $g$ and $g_c$ respectively and $g_m$ describes the interspecies coupling strength. For simplicity we assume equal masses $m$ for both components and perform numerical simulations by scaling interaction strengths in units of the BEC interaction strength $g_c$. In all figures presented in the paper lengths are therefore displayed in units of $x_0 = \hbar^2/mg_c$, time in units of $\omega^{-1} = \hbar^3/mg_c^2$ and energy in units of $\hbar\omega = mg_c^2/\hbar^2$. Choosing for example Rubidium-87 with a mass of $m \approx 87\,m_u$ and a reference value of $g_c = 7.7 \times 10^{-38}$ Jm $\approx 2\hbar\omega_\perp a_s$ corresponding to the intraspecies interaction of Rubidium-87 in a typical quasi-one-dimensional setup with a radial trapping frequency of $\omega_\perp \approx 2\pi \times 11$ kHz and a scattering length of $a_s \approx 100\,a_B$ [41] leads to $x_0 \approx 1\,\mu$m. In this work we extend the results of Ref. [10] and go beyond the Tonks-Girardeau

(TG) limit of $g \to \infty$ for the minority component by considering finite intraspecies repulsion $0 < g < \infty$. Consequently, the single-particle density of the immersed component is now calculated by tracing out all but one atom from the full many-particle wave function

$$\rho(x) \equiv |\Phi(x)|^2 = \int dx_2 \dots dx_N |\Phi(x, x_2, \dots, x_N)|^2 . \tag{2}$$

The condensate is assumed to be in free space with an average density $n_c \equiv N_c/L_c = \mu_0/g_c$ with $\mu_0$ describing the chemical potential of the condensate in the uncoupled case, whereas the immersed component is confined to a box potential of width $L$ with $V(x) \equiv 0$ for $|x| \leq L/2$ and $V(x) \equiv \infty$ otherwise.

## 3  Fermionization process

While in the TG limit of infinite intraspecies repulsion $g \to \infty$ the immersed component undergoes a transition to an insulating pinned state, for finite intraspecies interaction strengths the immersed gas can also exist in a cohesive superfluid state if the interspecies coupling $g_m$ is able to overcome the intraspecies repulsion $g$. In this section we study the superfluid state and show how the system fermionizes by transitioning to the pinned state as a function of $g$. For finite intraspecies interactions $g$, numerically solving the coupled equations (1) for more than a few atoms in the immersed component becomes challenging and we therefore restrict our consideration to the case of $N = 2$ and $N = 3$ immersed particles.

Starting from the limit of vanishing intraspecies interaction $g \to 0$ the immersed species can be effectively described by $N$ overlapping and identical single-particle wave functions and we refer to this cohesive state as the superfluid state. We describe this state using the effective model developed in Ref. [10] which is based on the Thomas-Fermi approximation for the condensate wave function in the regime of heavily imbalanced particle numbers as well as a weak interspecies interaction $g_m \ll \mu_0 L/N$. This leads to a description of the density of the immersed component in terms of bright solitons according to

$$\rho_{\mathrm{sf}}(x) = N \frac{a_{\mathrm{sf}}}{2} \frac{1}{\cosh^2(a_{\mathrm{sf}} x)} , \tag{3}$$

where the peak height $a_{\mathrm{sf}}$ is given by

$$a_{\mathrm{sf}}(g = 0) = N a_0 \frac{\sqrt{1 + 2N^2 \epsilon} - 1}{N^2 \epsilon} , \tag{4}$$

with $a_0 = m g_m^2/(2 g_c \hbar^2)$. The factor $\epsilon = 6 a_0^2 \hbar^2/(5 m \tilde{\mu})$ accounts for the energetic cost of deforming the BEC background with the modified chemical potential $\tilde{\mu} = \mu_0(1 + g_m N/g_c N_c)$. It is important to note that the closed expression for $a_{\mathrm{sf}}$ in Eq. (4) yields the best agreement with the numerically observed densities in the case of moderate interaction strengths $g_m$, where the width of the atomic wave function and the density dip in the condensate are proportional to each other, whereas for larger values of $g_m$ minimizing the energy functional for the total system energy provides a better match [10].

In order to obtain an approximation to the state of the system for finite $g > 0$, we use perturbation theory to calculate the contribution of the intraspecies interaction to the total energy. Following Eq. (3) we assume the wave function is a product of single particle solitons localized at the center of the trap

$$\Phi(\mathbf{x}) = \prod_{n=1}^{N} \sqrt{\frac{N a_{\mathrm{sf}}}{2}} \frac{1}{\cosh(a_{\mathrm{sf}} x_n)} , \tag{5}$$

and calculate the expectation value of this mean-field wave function according to

$$V_{\text{intra}} = g \sum_{k<l}^{N} \frac{\langle \Phi(\mathbf{x}) | \delta(|x_k - x_l|) | \Phi(\mathbf{x}) \rangle}{\langle \Phi(\mathbf{x}) | \Phi(\mathbf{x}) \rangle} = \frac{N}{2} (N-1) g \frac{a_{\text{sf}}^2}{4} \int dx \frac{1}{\cosh^4(a_{\text{sf}} x)}$$

$$= N (N-1) g \frac{a_{\text{sf}}}{6} . \tag{6}$$

Using the aforementioned Thomas-Fermi approximation $\psi(x,t) = \sqrt{(\tilde{\mu} - g_m |\Phi|^2)/g_c} e^{-i\tilde{\mu}t/\hbar}$ for the BEC wave function, the total system energy in the superfluid state reads

$$E_{\text{sf}} = N \left[ \frac{\hbar^2 g_m^2}{30 m \tilde{\mu} g_c} N a_{\text{sf}}^3 + \frac{\hbar^2}{m} \frac{a_{\text{sf}}^2}{6} - \frac{g_m^2}{6g_c} N a_{\text{sf}} + (N-1) g \frac{a_{\text{sf}}}{6} \right] + \frac{\tilde{\mu}^2 L_c}{2g_c} , \tag{7}$$

where the bright soliton parameter $a_{\text{sf}}$ minimizing the energy for finite $g > 0$ is now modified to

$$a_{\text{sf}} = -\frac{5\tilde{\mu}}{3} \frac{g_c}{N g_m^2} + \sqrt{\frac{5m\tilde{\mu}}{3\hbar^2} \left[ 1 - \frac{g_c g}{g_m^2} \left( 1 - \frac{1}{N} \right) \right] + \frac{25\tilde{\mu}^2}{9} \frac{g_c^2}{N^2 g_m^4}}$$

$$= N a_0 \frac{\sqrt{1 + 2N^2 \epsilon \left[ 1 - \frac{g_c g}{g_m^2} \left( 1 - \frac{1}{N} \right) \right]} - 1}{N^2 \epsilon} , \tag{8}$$

using the same parameter $\epsilon = 6a_0^2 \hbar^2/(5m\tilde{\mu})$ as before. This approximation only gives a valid inverse width $a_{\text{sf}} > 0$ for $g < g^* = N g_m^2/[(N-1)g_c]$.

Figure 1 shows the energy per immersed particle for increasing intraspecies interaction $g$ at fixed interspecies coupling (a) $g_m = 2g_c$ and (b) $g_m = 3g_c$. For small values of $g$, the numerical values (colored lines) agree very well with the analytical energy for the superfluid state from Eq. (7) (dot-dashed lines), both for $N = 2$ and $N = 3$, before starting to deviate and very slowly approaching the energy of the pinned state (dashed line) in the large $g$ limit. The pinned state is a system of individually isolated and localized single particle states which are separated from each other at particle positions $d_n$ and its density is described by

$$\rho_{\text{pin}}(x) = \frac{a_{\text{pin}}}{2} \sum_{n=1}^{N} \frac{1}{\cosh^2 \left( a_{\text{pin}}(x - d_n) \right)} , \tag{9}$$

with the peak height $a_{\text{pin}} = a_0 \left( \sqrt{1 + 2\epsilon} - 1 \right)/\epsilon$. For sufficiently large inter- and intraspecies interactions the overlap between particles vanishes and they arrange in a periodic structure with spacing $L/N$. The loss of coherence and the ordering of the particles has similarities to the Mott-insulating state in lattice systems [7], albeit in this case triggered by intercomponent interactions without the need for an external lattice potential. In this self-pinned state the system has a total energy of

$$E_{\text{pin}} = N \left( \frac{\hbar^2 g_m^2}{30 m \tilde{\mu} g_c} a_{\text{pin}}^3 + \frac{\hbar^2}{m} \frac{a_{\text{pin}}^2}{6} - \frac{g_m^2}{6g_c} a_{\text{pin}} \right) + \frac{\tilde{\mu}^2 L_c}{2g_c} , \tag{10}$$

and is independent of the interspecies interaction $g$ [10].

The insets of Fig. 1 show a close-up of the point where the numerically obtained energies approach their respective pinned state energies. Interestingly, our results suggest that the matter-wave trapping potential created by the background BEC enables a first-order phase transition in which the immersed component actually reaches the asymptotic state predicted

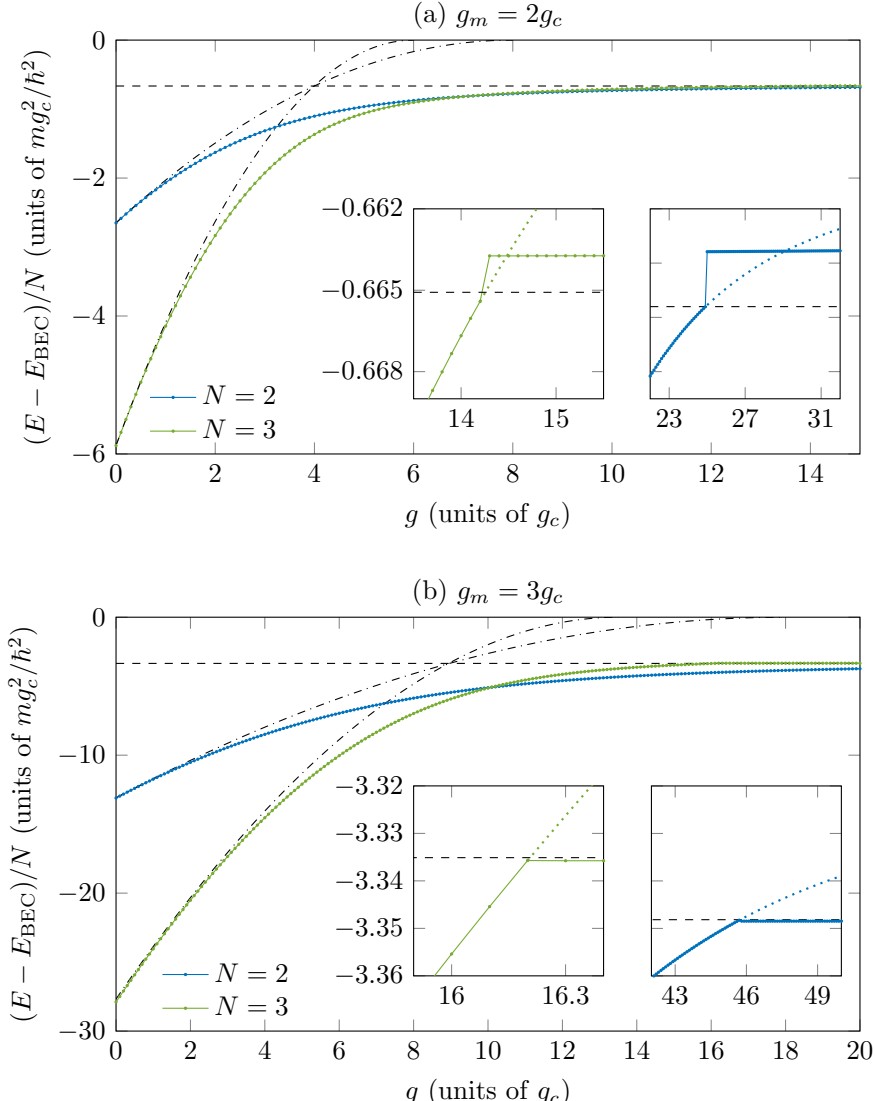

Figure 1: Total system energy per immersed particle, adjusted for $E_{\text{BEC}} = \tilde{\mu}^2 L_c / 2g_c$, as a function of the intraspecies interaction strength $g$ for a system of $N = 2$ (blue line) and $N = 3$ (green line) particles at fixed interspecies interaction strength $g_m = 2g_c$ (a) and $g_m = 3g_c$ (b) and density $N/L = 1/4 \ mg_c/\hbar^2$. The dashed line shows the analytical value for the pinned state energy $E_{\text{pin}} = -0.6656 \ mg_c^2/\hbar^2$ (a) and $E_{\text{pin}} = -3.3482 \ mg_c^2/\hbar^2$ (b) for $N = 2$ from Eq. (10), while the dashed-dotted lines show the energy per particle calculated from Eq. (7). The insets show a zoom of the phase transition point. The colored dotted lines indicate the energy of the metastable superfluid state if $g$ is increased beyond the transition point. The difference in background density of the BEC of $\mu_0 = 10^3 \ mg_c^2/\hbar^2$ ($N = 2$) and $\mu_0 = 2/3 \times 10^3 \ mg_c^2/\hbar^2$ ($N = 3$) leads to the slight difference in $E_{\text{pin}}$ visible in the insets.

in the TG limit $g \to \infty$. This is in contrast to the fermionization process in static trapping potentials, where the system only asymptotically approaches the corresponding energy level from below, but never reaches it, as can be seen for the analytically solvable case of two harmonically trapped atoms [42]. This behavior is also reminiscent of the discontinuous nature of the 1D superfluid-supersolid phase transition in dipolar condensates in the low-density limit

observed in numerical simulations [43] and recently confirmed experimentally [44]. In the thermodynamic limit however, the transition is predicted to be continuous in 1D systems and discontinuous in the 2D case [45]. As expected, the intraspecies repulsion in the bigger system of $N = 3$ is comparatively larger and leads to a lower value of $g$ at which the system crosses the transition to the pinned state.

Numerically the data was obtained via a self-consistent imaginary time evolution of the coupled system of Eqs. (1) using the Fourier split-step method [46]. We calculate the energy of the superfluid branch ($E_{sf}^{num}$) by starting from the non-interacting case $g = 0$ and adiabatically increasing the value of $g$ in small increments while using the previously found ground state as new initial state for the imaginary time evolution in each interaction step. In the insets of Fig. 1 the colored dotted lines indicate the energy of the then metastable superfluid state if $g$ is increased beyond the transition point. Similarly, for the pinned branch ($E_{pin}^{num}$) we start deep in the strongly interacting regime $g \ggg 1$ and use the ansatz from Ref. [10] as the initial state, adiabatically decreasing the value of $g$ in small decrements while again using the previously found ground state as new initial state for the imaginary time evolution. Finally, the combined physical branch is obtained from determining the minimum of both branches at each interaction step $E(g) = \min\{E_{sf}^{num}(g), E_{pin}^{num}(g)\}$.

In panel (a) of Fig. 1 the insets also show that the numerically determined energy of the pinned state is slightly larger than the dashed line obtained from the analytical expression Eq. (10). This is due to the fact that we are limited to studying a finite system numerically and that for small values of $g_m$ the resulting pinned states are also only weakly localized. The wave function of the immersed component is influenced by the box potential edges in that case, lifting the energy of the state in return. In contrast, for larger values of $g_m$ like in panel (b), the immersed particles are localized stronger, therefore not experiencing an influence of the system boundaries anymore, and the numerical values lie slightly below the analytical dashed line as expected from using the closed expression for $a_{pin}$ mentioned earlier compared to minimizing the complete energy functional. In order to minimize these numerical finite size effects on the results, we determine the phase transition point from the value of $g$ for which the numerically obtained energies of the superfluid state intersect with the analytically determined value for the pinned state according to Eq. (10), i.e. $E_{sf}^{num} = E_{pin}$, leading to the apparent unphysical discontinuity. As can be seen in the insets of Fig. 1, this has essentially no effect for larger values of $g_m$ while allowing us to obtain a clean and coherent phase transition line in the region of small $g_m$ where these finite size effects play a larger role as shown in the next section.

In order to obtain an estimate for the phase transition point according to our analytical model, we consider the limit $\tilde{\mu} \to \infty$ and $\epsilon \to 0$ in which we can neglect the energetic cost of deforming the BEC in Eqs. (10) and (7). In that case we have

$$a_{pin}(\epsilon \to 0) = a_0, \quad \text{and} \quad a_{sf}(\epsilon \to 0) = Na_0 - \frac{mg}{2\hbar^2}(N-1), \tag{11}$$

and the energy of the pinned state reduces to $E_{pin}(\epsilon \to 0) = -N\hbar^2 a_0^2/6m$, while the energy of the superfluid state becomes

$$E_{sf}(\epsilon \to 0) = N\frac{\hbar^2 a_{sf}^2}{6m} - \frac{a_0}{3}\left[1 - \left(1 - \frac{1}{N}\right)\frac{mg}{2a_0\hbar^2}\right]N^2 a_{sf}. \tag{12}$$

It is now easy to check that by choosing

$$g = \frac{\hbar^2}{m}2a_0 = \frac{g_m^2}{g_c}, \tag{13}$$

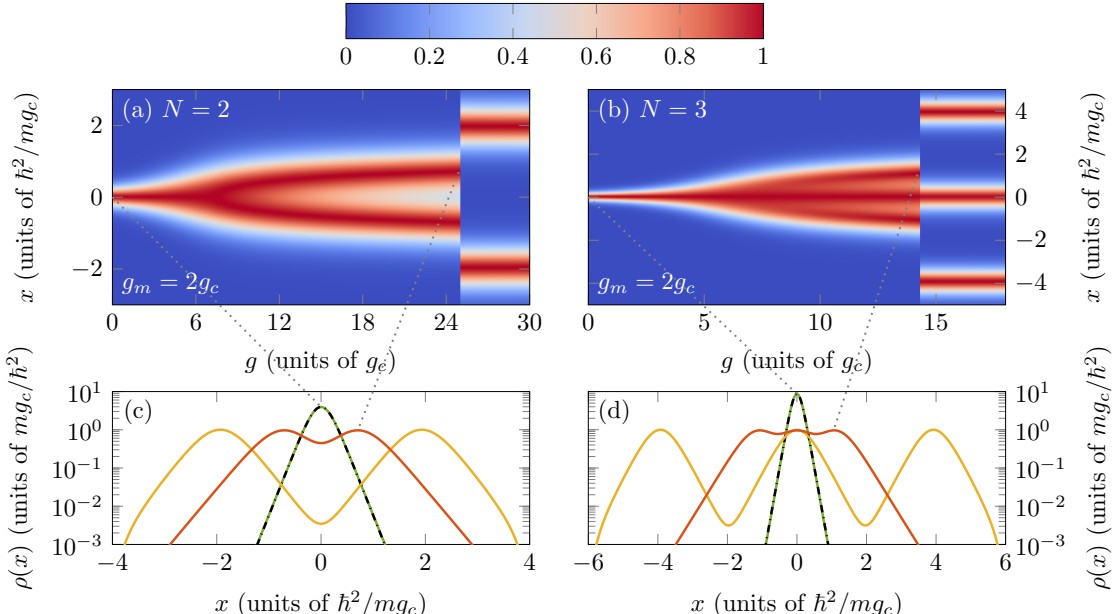

Figure 2: Top row: Renormalized minority component density $\rho(x)$ as a function of the intraspecies interaction strength $g$ for a system of (a) $N = 2$ and (b) $N = 3$ particles at fixed interspecies interaction strength $g_m = 2g_c$ and density $N/L = 1/4$ $mg_c/\hbar^2$. Bottom row: The corresponding line densities in the superfluid state at $g = 0$ (green line) and right before the transition (red lines) at $g_{num}^{crit} \simeq 24.9g_c$ and $g_{num}^{crit} \simeq 14.2g_c$ are shown to scale in (c) and (d) respectively. The yellow lines show the density in the pinned state for $g > g_{num}^{crit}$. The black dash-dotted lines show the analytical model according to Eq. (3). Other parameters are $\mu_0 = 10^3 \, mg_c^2/\hbar^2$ ($N = 2$), $\mu_0 = 2/3 \times 10^3 \, mg_c^2/\hbar^2$ ($N = 3$).

we further have that also $a_{sf}(\epsilon \to 0, g = 2\hbar^2 a_0/m) = a_0$ and that at this point the energies of the superfluid and pinned state are identical, $E_{sf} = E_{pin}$, indicating the point where the phase transition occurs according to our model. Rewriting the above choice of $g$ as

$$g_m^{crit} = \pm\sqrt{g_c g} \tag{14}$$

shows that the criterion coincides with the miscibility criterion for a two-component BEC [47] and also with a stability criterion for a Bose-Bose mixture derived by Cazalilla and Ho in the Luttinger liquid framework [29]. Remarkably, Eq. (14) does not depend on the number of particles $N$ and Fig. 1 shows that the intersection point of the analytic curves for $E_{sf}$ and $E_{pin}$ agrees very well with the predicted values of $g^{crit} = 4g_c$ for $g_m = 2g_c$ [see panel (a)] and $g^{crit} = 9g_c$ for $g_m = 3g_c$ [see panel (b)]. Even for the largest values of $|g_m| = 5g_c$ and correspondingly the largest values of $\epsilon$ considered in this paper, the actual intersection point for the model curves differs less than 5% from the analytical criterion, i.e. $g^{crit}(|g_m| = 5g_c) \approx 24g_c < 25g_c = g_m^2/g_c$.

For the values of $g_m$ shown in Fig. 1, there is a large discrepancy between the predicted and the numerically observed value of $g_{num}^{crit}$ however. This can be explained by studying the corresponding minority component densities at fixed interspecies coupling $g_m = 2g_c$, which are plotted in Fig. 2 for both (a) $N = 2$ and (b) $N = 3$. The density is renormalized to its respective maximum at each point for clarity. Starting from line densities at $g = 0$ that are well described by Eq. (3), as can be seen in panels (c) and (d), the densities begin to split into separate branches according to the number of particles and start forming depletions caused by the intraspecies repulsion, thereby deviating from the ansatz we use to calculate the energy contribution $V_{intra}$. This leads to the observed discrepancy between the analytical and

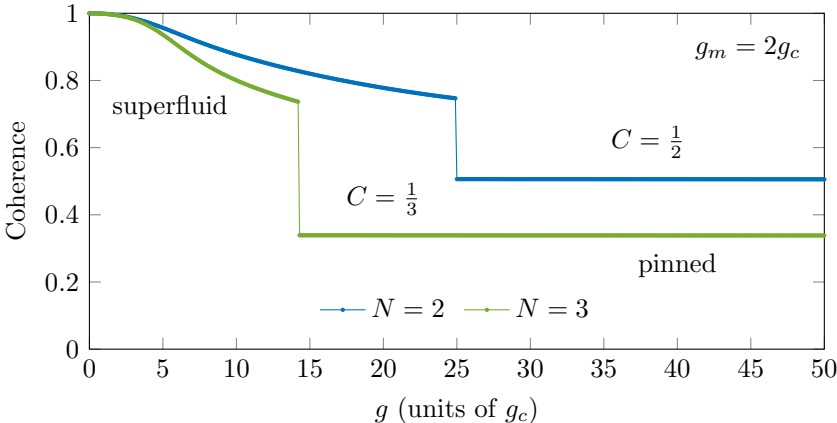

Figure 3: Coherence $C$ as a function of the intraspecies interaction strength $g$ for a system of $N = 2$ (blue line) and $N = 3$ (green line) particles at fixed interspecies interaction strength $g_m = 2g_c$ and density $N/L = 1/4 \, mg_c/\hbar^2$. Other parameters are $\mu_0 = 10^3 \, mg_c^2/\hbar^2$ ($N = 2$), $\mu_0 = 2/3 \times 10^3 \, mg_c^2/\hbar^2$ ($N = 3$).

numerical values of $g^{\text{crit}}$.

The branching occurs at comparatively lower values of $g$ for $N = 3$ particles than for $N = 2$ particles as one would expect from the larger contribution of the interaction energy in the case of more particles. At the numerically obtained critical values of $g_{\text{num}}^{\text{crit}} \simeq 24.9g_c$ ($N = 2$) and $g_{\text{num}}^{\text{crit}} \simeq 14.2g_c$ ($N = 3$) the pinned state becomes energetically favorable in both cases as can be seen in the insets of Fig. 1, resulting in an abrupt change towards a complete spatial separation of the atoms in the minority component. Panels (c) and (d) also show the densities right before and after the transition. The logarithmic scale used in the plots clearly demonstrates the difference in the central dip between the coherent superfluid state, which is still largely cohesive and centered around the origin, and the fully separated insulating pinned state.

## 4 Phase diagram

In order to distinguish the superfluid from the pinned phase and map out the phase diagram of the system, we calculate the coherence (also known as the condensate fraction) of the immersed component, $C = (\max_n \lambda_n)/N$. It characterizes the off-diagonal long-range order [48–50] and it is defined in terms of the largest eigenvalue $\lambda_n$ of the reduced single-particle density matrix (RSPDM), obtained from

$$\rho(x, x') = \int dx_2 \ldots dx_N \, \Phi^*(x, x_2, \ldots, x_N)\Phi(x', x_2, \ldots, x_N) = \sum_n \lambda_n \varphi_n^*(x)\varphi_n(x'), \quad (15)$$

with the natural orbitals $\varphi_n(x)$. The eigenvalues $\lambda_n$ describe how occupied the natural orbital states are and therefore are a good indication of the coherence in the system. For instance, for the superfluid state the lowest orbital is maximally occupied with $\lambda_0 \sim N$ which entails a maximal coherence of $C \to 1$. In the pinned state each particle is individually localized and isolated from one another, the result of which is the minimal coherence of $C \to 1/N$. The pinned state therefore resembles the reduced state of spinless fermions with $\lambda_n = 1$ for $n = 0, \ldots, N-1$ and $\lambda_{n \geq N} = 0$ as the contact interactions between the particles are nullified by the mean-field potential.

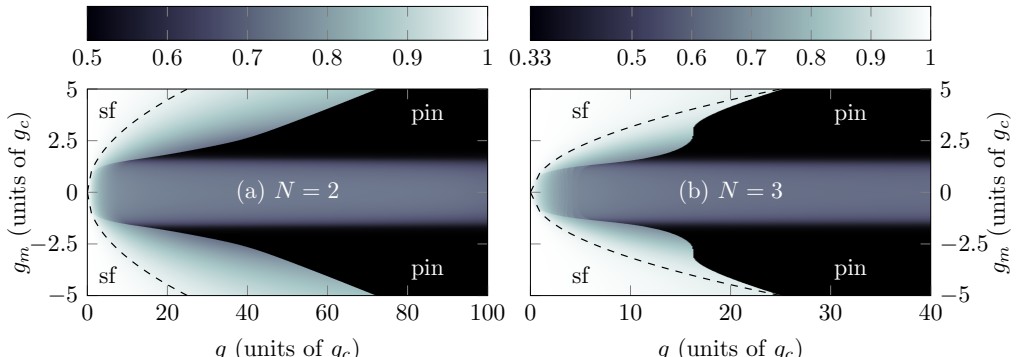

Figure 4: Phase diagram of the system for (a) $N = 2$ and (b) $N = 3$ as a function of intraspecies interaction $g$ and interspecies interaction $g_m$ for fixed density $N/L = 1/4$ $mg_c/\hbar^2$, exhibiting both a superfluid (sf) and pinned (pin) state of the immersed component. The colormap shows the value of the coherence $C$ while the dashed line indicates $g_m^2 = g_c g$. The gray shaded areas between $|g_m| \lesssim 1.5 g_c$ with $C > 1/N$ belong to the self-pinned state and are a result of the finite spatial system size, while the off-white areas in the superfluid phase stem from the finite particle number $N$ (see text for details). Other parameters are $\mu_0 = 10^3 \, mg_c^2/\hbar^2$ ($N = 2$), $\mu_0 = 2/3 \times 10^3$ $mg_c^2/\hbar^2$ ($N = 3$).

In Figure 3 we show the coherence as a function of intraspecies repulsion $g$ for fixed $g_m = 2g_c$ and fixed density $N/L = 1/4 \, mg_c/\hbar^2$. For $g = 0$ the coherence takes its maximum value for both the $N = 2$ and $N = 3$ particle systems, however for small finite values of $g$ the coherence is smaller than 1. This is a consequence of the RSPDM becoming mixed when interactions are finite which therefore reduces the coherence of the state and signifies the presence of quantum correlations between the particles. In fact, it is worth noting that in the Tonks-Girardeau limit $\lambda_0 \sim \sqrt{N}$ as $g \to \infty$ [48, 51] which still characterizes a superfluid-like phase that possesses some long range order. The coherence in the TG limit therefore scales as $C \to 1/\sqrt{N}$, and while this is distinct from a fully incoherent fermionic state with $C \to 1/N$, this is no longer true in the thermodynamic limit where both values vanish. Care must therefore be taken when considering large systems, however for the finite systems studied here this is not a concern. Indeed the coherence we obtain in the superfluid phase for $N = 2$ and $N = 3$ is always larger than $1/\sqrt{N}$ with the transition to the pinned phase signalled by a sudden decrease in coherence to $C = 1/N$.

In Figure 4 we show the full phase diagram in terms of the coherence for a system of (a) $N = 2$ and (b) $N = 3$ immersed particles as a function of both intraspecies interaction strength $g$ and interspecies interaction strength $g_m$. In general, for fixed intraspecies interaction $g$ the system transitions from the pinned insulating state to the coherent superfluid state if the interspecies interaction strength $|g_m|$ is increased until it can counteract the intrinsic repulsion of the immersed component. Similarly, for fixed interspecies coupling $g_m$ the system transitions from superfluid to pinned if the intraspecies repulsion $g$ is large enough so that the particles can push themselves apart. This means that for the self-pinning transition the system behaves contrarily to the pinning transition in an external lattice potential, where for finite values of $g$ the system is pinned for deep lattices but remains superfluid in shallow ones [6, 7].

For small values of $|g_m| \lesssim 1.5 g_c$ the phase diagrams both exhibit areas in the pinned region where $C > 1/N$. While this suggests the re-emergence of the superfluid phase it is in fact a result of finite size effects stemming from the particles hitting the edge of the box potential. In this regime the weak interaction with the BEC is not enough to isolate the particles. Instead, the trap edges keep the particles together, leading to larger values of the coherence. In this

region the finite system size also affects the numerically obtained values of the energy in the pinned state as detailed in the previous section and shown in the insets of Fig. 1. Therefore, in order to obtain a clean and coherent phase transition line in this region, we have determined the critical lines in both phase diagrams by checking where the numerically obtained ground state energy in the superfluid state $E_{\mathrm{sf}}^{\mathrm{num}}$ matches the analytical value of $E_{\mathrm{pin}}$ according to Eq. (10) as described earlier.

In general, the phase diagrams are symmetric with respect to the sign of the coupling parameter $g_m$, as was the case in Ref. [10]. This is a result of the homogeneous BEC density and is also reflected in the fact that our effective model only depends on $g_m^2$. For an inhomogeneous background gas the phase diagrams are not symmetric in general, as only attractive couplings $g_m < 0$ might lead to a stable trapped state while repulsive couplings $g_m > 0$ commonly result in phase separation in that case.

The miscibility criterion derived in Eq. (14) is also shown as a dashed line in both phase diagrams in Fig. 4. Particularly in the $N = 2$ case it underestimates the observed numerical value of $g_{\mathrm{num}}^{\mathrm{crit}}$ by a large amount due to the branching of the immersed component density described earlier. For $N = 3$ the behavior is similar in the region of small $g_m$ and small $g$. However, for $g \gtrsim 15 g_c$ a distinct shoulder appears in the phase transition line, drawing it closer to the analytical estimate. In this region, the larger interspecies coupling leads to a strong localization of the immersed component and the formation of the density modulation due to the intraspecies repulsion, visible in Fig. 2, is suppressed. In other words, the numerically obtained critical point is close to the prediction Eq. (14) when the line density is still well described by our bright solitonic ansatz.

We expect that in the thermodynamic limit $N \to \infty$ the phase transition line will converge to our analytical estimate in the region of small $g_m$, leading to a disappearance of the off-white regions seen in the superfluid phase between the dashed approximation and the actual transition lines as a result of the finite number $N$ of immersed particles. For larger $g_m$, the effect of $\epsilon > 0$, which we have neglected in the derivation, becomes relevant, leading to a transition from superfluid to pinned at weaker intraspecies repulsion than predicted. This is also in line with and even necessary for our observation in Ref. [10] that in the TG limit $g \to \infty$ only the pinned state exists, as the BEC is not able to compress the fermionized bosons for any value of the interaction strength $g_m$.

## 5 Conclusion

We have studied how a small, initially superfluid, one-dimensional Bose gas immersed into a Bose-Einstein condensate fermionizes as a function of increasing intraspecies repulsion $g$ and eventually reaches the insulating self-pinned state expected in the Tonks-Girardeau limit $g \to \infty$. In contrast to static trapping potentials, this asymptotic state is actually crossed and not just approached from below, implying a first-order phase transition as a result of the matter-wave backaction. We have confirmed this behavior by numerically simulating the system for $N = 2$ and $N = 3$ immersed particles and calculating its density, energy and coherence. We have extended the effective model presented in Ref. [10] to the superfluid state and used it to derive a phase transition line valid in the mean-field limit $N \to \infty$ and $\epsilon \to 0$, i.e. $N_c \to \infty$. Finally, we mapped out the phase diagram as a function of interspecies and intraspecies couplings with extensive simulations for the aforementioned $N = 2$ and $N = 3$ cases.

Regarding future work, it would be interesting to probe the dynamics of the system by quenching or ramping either the intraspecies or interspecies interaction strengths across the phase transition lines. Furthermore, while we are limited to few-body systems due to numerical restrictions, it would be beneficial to study the system for a larger number of immersed

particles $N$ in order to investigate how the transition from superfluid to pinned scales with system size. This will also allow putting the effective model to the test and to extend the phase diagram to finite temperatures by studying the stability of the superfluid phase against thermal excitations, similar to Ref. [10]. It is also worthwhile to explore the role of finite size effects on the phase diagram, particularly by considering periodic boundary conditions for both the immersed gas and the BEC. Already calculating the phase diagram for $N = 3$ particles with the Fourier split-step method using $N_{\mathrm{grid}} = 512$ position grid points at the resolution in $g$ and $g_m$ presented in Fig. 4 (b) required on the order of $10^6$ CPU hours, despite numerous optimizations. Larger systems therefore need to be studied by alternative techniques such as world-line [52, 53] and diffusion Monte-Carlo methods [54, 55], multi-configurational time dependent Hartree methods [56], or the density-matrix renormalization group (DMRG) [57] and particularly its continuum extension [58]. All of these techniques are also able to cover the finite temperature case and specifically the diffusion Monte-Carlo technique has already been used for studying 1D droplets in binary Bose mixtures [59, 60].

## Acknowledgements

The authors would like to thank Mathias Mikkelsen and Tai Tran for fruitful discussions and for performing supplementary simulations.

**Funding information** This work has been supported by the Okinawa Institute of Science and Technology Graduate University and used the computing resources of the Scientific Computing and Data Analysis section. T. K. acknowledges support from a Research Fellowship for Young Scientists by the Japan Society for the Promotion of Science under JSPS KAKENHI Grant No. 21J10521. T. F. acknowledges support from JSPS KAKENHI Grant Number JP23K03290. T. F. and T. B. are also supported by JST Grant Number JPMJPF2221.

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
