# Peer review of "Fermionization of a Few-Body Bose System Immersed into a Bose-Einstein Condensate"

_SciPost Physics, doi:SciPost Phys. 15, 095 (2023)_

## Round 1 · Referee Report · Anonymous (Referee 1) · 2023-3-17

Strengths

1 The system being investigated is relevant to recent experiments and has the potential to be realized using present experimental techniques
2 A quantum phase transition has been predicted at zero temperature

Weaknesses

1 -Certain figures may be misleading, such as the energy of a metastable state appearing lower than the ground-state energy in certain regimes, while the opposite is expected
2 -The article's clarity would improve if dimensionfull units were utilized
3 -In the considered case of few-particle systems comprising only two or three atoms in the second component, the terminology of "phase transitions" requires clarification

Report

The Authors study a two-component bosonic system in a one-dimensional geometry. Physically, the considered situation corresponds to having one component in the mean-field Gross-Pitaevskii regime, while the second component is assumed to have arbitrary interaction strength with the first component, and each component has repulsive intraspecies repulsion. It is found that the considered model manifests a first-order phase transition from a superfluid to normal (pinned) phase.

While potentially the article might be appropriate for an eventual publication in SciPost a number of issues should be settled down first.

Requested changes

1- The minority particles are confined to a hard-walled box. This typically induces a finite-size effect in the energy of the order of $1/N$. Instead using periodic boundary conditions seems to be much more natural as (a) finite-size effects would scale much more favorably, as $1/N^2$ (b) the geometry will be much more similar to that which appears in the thermodynamic limit, $N\to\infty$. Is it possible to use periodic boundary conditions for the minority component? 2- Quantum phase transitions are predicted for $N=2$ and 3 particles in the minority component. The concept of a phase transition is properly defined in the thermodynamic limit. Please justify why the same terminology applies here and in which sense it should be understood. 3- While the "pinned" term is widely used in the article, pinning makes a reference to some dynamic quantity, like the inability of the particles to move. Also having zero winding number or a gap in the excitation spectrum would mean that the system is no longer superfluid. Please explain better in which sense the system is pinned. 4- It would be useful to mention phase separation in the Abstract 5- In the Introduction, the pinning transition is discussed in the context of an optical lattice, but it also occurs in the presence of a random external field. The Renormalization Group theory predicts the critical values of the Luttinger parameter where the corresponding pinning transitions occur. This can be mentioned 6- It is tempting to relate the immediate change of phase in the Tonks-Girardeau gas as a manifestation of the orthogonality catastrophe given that the system is equivalent to ideal fermions. 7- Introduction "the self-bound droplets are also predicted to have a bright solitonic shape". Clarify the meaning of a "bright soliton" as intended here. 8-Authors opted for using dimensionless parameters without explicitly relating them to the full units. As a result, it is not always obvious how to get back to the full units, for example, the peak height $a_{sf}$ and $N/L = 1/4$ probably should have units of density, etc. I suggest introducing the model in Eqs (1a-1b) using full units, including $\hbar$ and $m$, and using full units in the figures. 9- Fig. 1, add labels to the vertical and horizontal axes. Is that $E / E_{BEC}$? 10- is Thomas Fermi approximation equivalent to the Local Density Approximation here? If yes, it might be useful to mention the other name. 11- It is not clear why some specific values of parameters are used. For example, there are figures for $g_m=2$ and $g_m=3$ but not $g_m=1$, also the case of negative values of $g_m$ can be also reported. Why $N/L = 1/4$ is used? 12- I am somehow confused by the "energy per particle" label, the total energy should contain $N=2 -3$ particles in the minority component and an infinite number of particles in the majority component. In the thermodynamic limit, the effect of the minority component is negligible on the total energy. So what is actually meant by the "Energy per particle"?

Minor Comments

13- there are three coupling constants in the problem, which are labeled as $g$, $g_c$, and $g_m$. Please give a hint why such labels are used. Alternatively, different components can be labeled as A and B or $\uparrow$ and $\downarrow$. 14- "few body gas" might sound strange 15- there is no need of introducing $N$ in the Abstract 16- "We present an analytical model that includes the superfluid state ..." please rephrase 17- abbreviation "1D" is used without being introduced 18- "Associating `pseudospin' $\frac{1}{2}$ with such systems...". Associating a pseudospin with a system sounds strange. Please rephrase 19- "the ground-state of such interacting bosons"->"the ground state of such interacting bosons" 20- remove ident after Eq. (1b) 21- It seems that $\mu_0$ is used without being introduced 22- There is no consistency in using notation $x/y$ or $\frac{x}{y}$ in the text of the Article. Please check which one is allowed or preferred by the journal and switch to using it. 23- Change in the source code Eq. () to Eq.~() so that the number and the equation always stay in the same line 24- "Fig. 1" at the beginning of a sentence sometimes has to be expanded into "Figure 1". Please check if this rule also applies in this journal. 25- "evolution of the coupled system using ...", add an explicit reference to the used equation 26- Figs 2(a,b) use linear scale while (c,d) a logarithmic scale. Would it be better to use also a linear scale in (c,d)? 27- It seems to me that the definition of the "coherence $C$" is nothing else but the condensate fraction in its original definition. Condensate fraction is a much wider used term. 28- Phase diagram, Fig 4, the labels contain two phases, "SF" and "pinned", while the figure has four regimes, as shown by a dashed line and white, black, and gray colors. It is not clear from the figure how the gray and right-white regions are classified.

  • validity: high
  • significance: high
  • originality: high
  • clarity: good
  • formatting: excellent
  • grammar: good

Author:  Tim Keller  on 2023-05-08  [id 3648]

(in reply to Report 1 on 2023-03-17)

We thank the referee for their positive review of our article and their recommendation for publication after revision.

1 - The minority particles are confined to a hard-walled box. This typically induces a finite-size effect in the energy of the order of $1/N$. Instead using periodic boundary conditions seems to be much more natural as (a) finite-size effects would scale much more favorably, as $1/N^2$ (b) the geometry will be much more similar to that which appears in the thermodynamic limit, $N\rightarrow\infty$. Is it possible to use periodic boundary conditions for the minority component?

Response: Since in the infinite limit described by periodic boundary conditions the pinned state is only pinned with respect to each particle's neighbour, the appearance of the pinned state requires spontaneous symmetry breaking. This is difficult to simulate and we have therefore opted to rather break the symmetry by hand by introducing a box potential. This also significantly reduces the numerical complexity of the problem and allows to more accurately pinpoint the transition from superfluid to pinned.

2 - Quantum phase transitions are predicted for $N=2$ and 3 particles in the minority component. The concept of a phase transition is properly defined in the thermodynamic limit. Please justify why the same terminology applies here and in which sense it should be understood.

Response: The referee is correct that the pinning transition in our work only consists of a couple of particles in the minority component and we do not make any statement about the thermodynamic limit. What we show in our work is a sudden structural change in the particles density due to the back-action from the majority component. This is reminiscent of the superfluid to Mott-insulator transition in 1D lattice systems, where the lattice depth is replaced by the interspecies interaction strength, a connection we draw from the phase diagrams in Fig. 4. From these results and others in our work this suggests the emergence of an analogous superfluid to insulator transition in this system for a finite intraspecies interaction away from the Tonks-Girardeau limit. Unfortunately, using our current methods we are limited numerically to how large a system we can simulate and cannot investigate how this transition scales with system size. However, different techniques could be employed for continuous systems, such as the multi-configurational time dependent Hartree method [Cao et al., J. Chem. Phys. 147, 044106 (2017).] and exact diagonalization of the full system [Garcia-March et al., Phys. Rev. A 87, 063633 (2013)] and is something we are exploring. In the main text we have clarified that our work suggests a superfluid to pinned transition for finite intraspecies interactions and in the conclusions have added a brief discussion on extending this work to larger systems.

3 - While the "pinned" term is widely used in the article, pinning makes a reference to some dynamic quantity, like the inability of the particles to move. Also having zero winding number or a gap in the excitation spectrum would mean that the system is no longer superfluid. Please explain better in which sense the system is pinned.

Response: The referee is correct in that the wording 'pinned state' is not fully capturing the situation in the thermodynamic limit, as there is no external potential in which the gas is pinned. In the box potential the distance between the non-overlapping atoms is given by the Fermi momentum, which means that higher densities lead to pinned states with smaller lattice constants. The atoms are therefore pinned with respect to each other. We have updated the main text to more carefully clarify what we mean by 'pinned'. There also exists a finite gap in the energy spectrum signalling the insulating phase as we have shown in our previous work Keller et al., PRL 128, 053401 (2022), and the pinned state possesses no many-body coherence as the particles do not overlap. The dynamical properties of this system is the focus of our ongoing work, with the mean-field lattice possessing unique characteristics compared to the static lattice potential.

4 - It would be useful to mention phase separation in the Abstract

Response: We have adjusted the abstract accordingly.

5 - In the Introduction, the pinning transition is discussed in the context of an optical lattice, but it also occurs in the presence of a random external field. The Renormalization Group theory predicts the critical values of the Luttinger parameter where the corresponding pinning transitions occur. This can be mentioned

Response: We thank the referee for this suggestion and have added a sentence and two references to the introduction.

6 - It is tempting to relate the immediate change of phase in the Tonks-Girardeau gas as a manifestation of the orthogonality catastrophe given that the system is equivalent to ideal fermions.

Response: While the pinned state is certainly rather orthogonal to the superfluid state, we do not see a direct connection with the orthogonality catastrophe. However, it would certainly be interesting to investigate this in a dynamical study in the future work.

8 - Introduction "the self-bound droplets are also predicted to have a bright solitonic shape". Clarify the meaning of a "bright soliton" as intended here.

Response: We have added a clarification describing bright solitons as structures with a density $\sim \cosh^{-2}(x)$.

9 - Authors opted for using dimensionless parameters without explicitly relating them to the full units. As a result, it is not always obvious how to get back to the full units, for example, the peak height $a_\mathrm{sf}$ and $N/L=1/4$ and probably should have units of density, etc. I suggest introducing the model in Eqs (1a-1b) using full units, including $\hbar$ and $m$, and using full units in the figures.

Response: We thank the referee for addressing the issue of relating the dimensionless quantities used in our manuscript to SI units. In general, for equations one can use dimensional analysis to obtain the corresponding expressions in SI units. For example, the peak height $a_\mathrm{sf}$ at $g=0$ expressed as a function of the coupling strengths (neglecting the effect of $\epsilon$) becomes

$$ a_\mathrm{sf} = N\frac{g_m^2}{2g_c} \longrightarrow \left(a_\mathrm{sf} = N \frac{g_m^2}{2g_c}\frac{m_\mathrm{TG}}{\hbar^2}\right)_{\mathrm{SI}} \; , $$
where $m_\mathrm{TG}$ denotes the mass of the immersed component. For quantities like length or energy one furthermore has to pick a reference scale. In our case the physical value $(g_c)_\mathrm{SI}$ of the BEC intraspecies interaction, which is already set to $g_c = 1$ throughout the manuscript, lends itself to that purpose. For a quick conversion between dimensionless and full units this can equivalently be thought of as measuring lengths in units of
$$ \left(x_0 = \frac{\hbar^2}{m_\mathrm{TG}g_c}\right)_\mathrm{SI} \; , $$
time in units of
$$ \left(\omega^{-1} = \frac{\hbar^3}{m_\mathrm{TG}g_c^2}\right)_\mathrm{SI} \; , $$
energy in units of
$$ \left(E_0 = \hbar\omega = m_\mathrm{TG}g_c^2/\hbar^2\right)_\mathrm{SI} $$
and interaction strengths in units of $(g_c)_\mathrm{SI}$.

While this also introduces a mass ratio $m_\mathrm{TG}/m_\mathrm{BEC}$ in the kinetic term of the BEC's Gross-Pitaevskii equation, this ratio can be neglected in the Thomas-Fermi regime considered throughout the manuscript. Choosing a typical value of

$$ \left(g_c\right)\mathrm{SI} \approx 2\hbar\omega\perp a_s \approx 7.7 \times 10^{-38} \text{ Jm} $$
corresponding to a quasi-one-dimensional gas with a radial trapping frequency of $\omega_\perp = 2\pi\times 11$ kHz and the scattering length of $a_s \approx 100$ $a_\mathrm{B}$ for Rubidium-87 with a mass of $m_{\mathrm{TG}}\approx 87$ $m_u$, leads to $x_0 \approx 1$ $\mu\mathrm{m}$ and $\omega^{-1} \approx 1.4$ ms and energy measured in units of $E_0/(2\pi\hbar)\approx 116.6$ Hz. This makes it possible to immediately relate the quantities plotted in the manuscript to physical units. We have added this explanation to the main text, but we believe that reintroducing full units to the equations and figures as suggested by the referee might add unnecessary complexity.

9 - Fig. 1, add labels to the vertical and horizontal axes. Is that $E/E_\mathrm{BEC}$?

Response: The plotted quantity is $(E-E_\mathrm{BEC})/N$ and we have changed the axis label in Fig. 1 accordingly for clarity (see also answer to question (12) below). If the referee's comment is referring to the insets of Fig. 1, they simply show close-ups of the phase transition point. Therefore the axis labels are identical to the main figure and we have omitted them for better readability.

10 - is Thomas Fermi approximation equivalent to the Local Density Approximation here? If yes, it might be useful to mention the other name.

Response: The Local Density Approximation refers to the assumption of a piece-wise constant potential, while the Thomas-Fermi approximation assumes that the kinetic energy of the BEC can be neglected, i.e. the BEC density is locally constant. While this implies that they are equivalent and it is tempting to make the connection, we believe that to avoid confusion it would be beneficial to stick with the Thomas-Fermi approximation as is more commonly used term in the cold atom community.

11 - It is not clear why some specific values of parameters are used. For example, there are figures for $g_m=2$ and $g_m=3$ but not $g_m=1$, also the case of negative values of $g_m$ can be also reported. Why $N/L = 1/4$ is used?

Response: We thank the referee for this question. The values of $g_m$ in Fig. 1 were chosen to depict representative behaviour of the system in different regions of the phase diagram, in particular with respect to the occurrence of finite size effects. For example, for values of $g_m=2$ and below, the energy of the system is still affected by the finite size of the numerically simulated box potential as explained in the main text, whereas for values of $g_m=3$ and above there is no noticeable effect of the finite box size. For negative values of $g_m$ the same density profiles in Fig.2 will be found for $g_m=-2$ and $g_m=-3$ and are therefore not shown. Similarly, for the value of $N/L$, i.e. the value of $1/L$, there is a trade-off between setting it as small as possible in order to minimize the effects of the finite box of length $L$, and setting it large enough to retain a spatial resolution of the box that is high enough for reliably distinguishing the small energy differences between the superfluid and self-pinned phase. Within these considerations our particular choice of $N/L = 1/4$ is mostly arbitrary, but was found to give consistent results based on the fixed BEC interaction strength $g_c=1$.

12 - I am somehow confused by the "energy per particle" label, the total energy should contain $N=2-3$ particles in the minority component and an infinite number of particles in the majority component. In the thermodynamic limit, the effect of the minority component is negligible on the total energy. So what is actually meant by the "Energy per particle"?

Response: The energy plotted in Fig.1 is the total energy $E$ of the system from which the bare energy of the BEC, which is several orders of magnitude larger than the other components, has been subtracted to give a sensible energy scale and the resulting quantity is divided by the number $N$ of immersed atoms. The resulting energy which is plotted in Fig.1 is then $(E-E_\mathrm{BEC})/N$ with $E_\mathrm{BEC}=\tilde{\mu}^2L_c/2g_c$. This representation also allows us to characterize states that are confined by the BEC as they will possess negative energy in this scale. Otherwise, states with positive energy are not confined in the BEC, but are rather trapped in the box potential. We have edited the axes of Fig.1 and the caption to make this more clear.

13 - there are three coupling constants in the problem, which are labeled as $g$,$g_c$, and $g_m$. Please give a hint why such labels are used. Alternatively, different components can be labeled as A and B or $\uparrow$ and $\downarrow$.

Response: We use this particular notion in order to be consistent with our previous publication in Ref. [8], where $g_m$ ('mix') describes the interspecies coupling between the immersed component and the background Bose-Einstein condensate, which has an intraspecies interaction strength of $g_c$ ('condensate'). In the present manuscript $g$ is the newly introduced intraspecies interaction strength of the immersed component that was only considered in the Tonks-Girardeau limit $g\rightarrow\infty$ in the previous publication. We have decided against using a notation like A and B or $\uparrow$ and $\downarrow$ commonly found in descriptions of two-component systems since these notations usually describe two equally sized components, whereas in our case we consider a heavily imbalanced system in which the focus is on the minority component immersed into the background Bose-Einstein condensate which only plays a subsidiary role. Moreover, since the particular value of $g_c$ does not qualitatively affect our results, its definition is only necessary for completeness.

14 - "few body gas" might sound strange

Response: We have changed it to "few-body system"

15 - there is no need of introducing $N$ in the Abstract

Response: We have changed the abstract accordingly.

16 - "We present an analytical model that includes the superfluid state ..." please rephrase

Response: We have reworded the corresponding sentence in the abstract.

17 - abbreviation "1D" is used without being introduced

Response: We thank the referee for the observation and have added a definition of the abbreviation.

18 - "Associating `pseudospin' $\frac{1}{2}$ with such systems...". Associating a pseudospin with a system sounds strange. Please rephrase

Response: We thank the referee for the suggestion and have rephrased the sentence.

19 - "the ground-state of such interacting bosons"->"the ground state of such interacting bosons"

Response: We have removed the hyphenation.

20 - remove ident after Eq. (1b)

Response: We have removed the indent.

21 - It seems that $\mu_0$ is used without being introduced

Response: The quantity $\mu_0$ is implicitly defined just below Eq. (2) and we have added an explicit mention for clarity.

22 - There is no consistency in using notation $x/y$ or $\frac{x}{y}$ in the text of the Article. Please check which one is allowed or preferred by the journal and switch to using it.

Response: We thank the referee for pointing this out and while we could not determine if there is a preferred style for the journal, we have switched to a consistent usage of $x/y$ for inline expressions.

23 - Change in the source code Eq. () to Eq.$\sim$() so that the number and the equation always stay in the same line

Response: We have fixed the source code.

24 - "Fig. 1" at the beginning of a sentence sometimes has to be expanded into "Figure 1". Please check if this rule also applies in this journal.

Response: We could not confirm if this rule applies for SciPost Physics, but we have expanded "Fig." to "Figure" if it occurs at the beginning of a sentence following the referee's suggestion.

25 - "evolution of the coupled system using ...", add an explicit reference to the used equation

Response: We have added a reference to the coupled system of Eqs. (1).

26 - Figs 2(a,b) use linear scale while (c,d) a logarithmic scale. Would it be better to use also a linear scale in (c,d)?

Response: We thank the referee for the observation, however the authors believe that the logarithmic scale shown in panels (c) and (d) is necessary to clearly distinguish the difference in density around $x=0$ between the superfluid state directly before the transition and the self-pinned state after the transition. On a linear scale they would seem nearly identical apart from the peak positions.

27 - It seems to me that the definition of the "coherence $C$" is nothing else but the condensate fraction in its original definition. Condensate fraction is a much wider used term.

Response: We thank the referee for pointing this out. However, we believe that for the purposes of the paper the term 'coherence' is better suited as the defining feature to distinguish the superfluid from the self-pinned phase of the immersed component. This is commonly used in the literature when discussing the reduced state of correlated systems, for instance see Colcelli et al, Phys. Rev. A 98, 063633 (2018) and Sowiński and García-March, Rep. Prog. Phys. 82 104401 (2019). Moreover, since we are considering a quasi-one-dimensional system the term condensate fraction might be misleading and also lead to confusion with regards to the background Bose-Einstein condensate.

28 - Phase diagram, Fig 4, the labels contain two phases, "SF" and "pinned", while the figure has four regimes, as shown by a dashed line and white, black, and gray colors. It is not clear from the figure how the gray and right-white regions are classified.

Response: The system exhibits two distinct phases, superfluid and pinned, and the two additional regimes appearing in the phase diagrams are a result of the finite system size, both in terms of spatial extent and particle number amenable to the numerical simulations. In the gray shaded area within $|g_m|\lesssim 1.5$ the system is also in the self-pinned state, however the comparatively weak interspecies interaction is not enough to fully localize the particles. As detailed in the main text, the increase in coherence over the expected value of $1/N$ is a result of the wave function hitting the edge of the box potential in that case. The off-white regions are in the superfluid state. The reduced coherence in these areas is a result of the finite particle number and the branching out of the immersed component density that is visible in Fig. 2. We expect the off-white region to vanish in the limit of large particle numbers $N\rightarrow\infty$ as can be seen already for the increase from $N=2$ to $N=3$ and as described in the main text. Lastly, the dashed line indicates the phase separation criteria in the thermodynamic limit and is added for comparison. We have expanded on these explanations both in the caption of Fig. 4 and in the main text.

---

## Round 1 · Referee Report · Andrea Richaud (Referee 2) · 2023-3-20

Strengths

1) Interesting new first-order quantum phase transition; 2) Motivation and relevance for ongoing research are stated very clearly; 3) Good combination of analytical and numerical techniques; 4) Extensive and careful analysis, encompassing several indicators; 5) Effective presentation of the results;

Weaknesses

1) Some terms seem to be used without prior explanation; 2) Analysis restricted to few immersed atoms ($N=2$ and $N=3$), even if larger values of $N$ would lead to a remarkable computational complexity; 3) Finite-size effects in the phase-diagram.

Report

The Authors present an interesting first-order quantum phase transition where some impurity atoms immersed in a quasi-1D BEC undergo the superfluid - insulator transition upon varying the repulsion between impurity atoms. The first-order character of this transition is explained to be ascribed to the matter-wave backaction.

The analysis is carried out both within an analytical variational approach and by means of extensive numerical simulations. Several indicators are computed and presented as suitable to support the discussion.

In my opinion, the general motivation is stated clearly, and the results are presented with due clarity. For these reasons, I believe that this work may be of interest for the community of quasi-1D Bose-Bose mixtures, also because the presented phenomenology should be experimentally accessible. I therefore recommend it for publication.

Nevertheless, I have some minor comments which the Authors may want to address before re-submission. They are intended to possibly make some discussions more clear and self-readable.

Requested changes

1- Eq. (5): Why is it reasonable to assume a "separable" wave function? A many-body wavefunction for identical bosons should be symmetrized.

2- Eq. (9): Was this equation derived in Ref. [8]? If so, I believe that a brief paragraph explaining how it was derived would make the current article more self-readable. Moreover: is this quantity referred to state whose $\rho(x)$ has only 1 peak or $N$ separated peaks?

3- In my opinion the term "pinned" is sometimes used without due explanation. What does it mean that a state of the type (5), i.e. associated to a single-peak $\rho(x)$, is "pinned"? I would naively think that, tilting the system, the immersed atoms would not move. Is this correct? In any case, I feel that an explanation is necessary.

4- When discussing some examples of first-order quantum phase transitions, the Authors may consider citing the following paper "Pathway toward the formation of supermixed states in ultracold boson mixtures loaded in ring lattices, Phys. Rev. A 100, 013609 (2019)" where the phase separation of an imbalanced quasi-1D Bose-Bose mixture is described to be either a first-order or a second-order quantum phase transition.

5- Fig. 1, panel (a), insets: both the solid green and the solid blue lines seem to be discontinuous functions (jump-like discontinuity). If I understood correctly, the energy per particle should be a continuous function of $g$, while its first derivative is discontinuous. Please briefly explain this in the main text.

6- In Fig. 2, the Authors plot $\rho(x)$. It may be worth plotting also the corresponding background densities $|\psi(x)|^2$.

7- Fig. 3: I would add the labels "1/2" and "1/3" to the two horizontal segments.

8- Concerning the description of $\rho(x)$ as a function of $g$ (see, e.g. Fig. 2), and the associated coherence properties, the authors never mention the concept of Mott-insulating state. Is there a reason why the authors omit this term? I think that, if it is correct, it may help the reader to understand the phenomenology at stake in a very effective way.

9- Concerning the finite-size effects in the phase diagram, the Authors say " it is in fact a result of finite size effects stemming from the particles hitting the edge of the box potential". Would periodic boundary conditions improve the result? I understand that it may be computationally more difficult to implement, but, I think that a brief comment about the difference open/periodic boundary conditions would be useful.

10- Fig. 2: Can the authors comment on the spacing between the $N$ peaks visible at high values of $g$?

  • validity: high
  • significance: high
  • originality: high
  • clarity: high
  • formatting: excellent
  • grammar: excellent

Author:  Tim Keller  on 2023-05-08  [id 3647]

(in reply to Report 2 by Andrea Richaud on 2023-03-20)

We thank the referee for their positive review of our article and their recommendation for publication.

1 - Eq. (5): Why is it reasonable to assume a "separable" wave function? A many-body wavefunction for identical bosons should be symmetrized.

Response: Our ansatz for the superfluid state assumes that the system is uncorrelated, such that we can describe it as a product state of identical single particle functions. We have rewritten the sentence to clearly state this.

2 - Eq. (9): Was this equation derived in Ref. [8]? If so, I believe that a brief paragraph explaining how it was derived would make the current article more self-readable. Moreover: is this quantity referred to state whose $\rho(x)$ has only 1 peak or $N$ separated peaks?

Response: The energy in Eq. (9) refers to the 'pinned' state where $N$ separated peaks are individually localized and isolated from one another. The referee is correct and the energy of the pinned state was derived in our previous work Ref. [8]. Following the referee's suggestion, we have added a brief reminder about how this quantity is derived. In our work the 'pinned' state refers to a situation where $N$ separated peaks are individually localized and isolated from one another. We have chosen this terminology in analogy to the more well-known pinned state that is observed when a Tonks-Girardeau gas is trapped in an external lattice potential with commensurate particle number and where the particle positions are given by the lattice periodicity. However, in our coupled two-component system the periodicity is instead given by $N/L$. In contrast, the superfluid state corresponds to the situation where the immersed particles have overlapping densities, i.e. only one peak is visible in $\rho(x)$. In this regime the intercomponent interactions confine the particles inside the BEC and they are not individually localized. We have rewritten the text more clearly to define what we consider a 'pinned' or 'superfluid' state in terms of the density.

3 - In my opinion the term "pinned" is sometimes used without due explanation. What does it mean that a state of the type (5), i.e. associated to a single-peak $\rho(x)$, is "pinned"? I would naively think that, tilting the system, the immersed atoms would not move. Is this correct? In any case, I feel that an explanation is necessary.

Response: The referee is correct in that the wording 'pinned state' is not fully capturing the situation in the thermodynamic limit, as there is no external potential in which the gas is pinned. In the box potential the distance between the non-overlapping atoms is given by the Fermi momentum, which means that higher densities lead to pinned states with smaller lattice constants. Furthermore, in such a finite potential a tilting of the box would not lead to a movement of the particles. However, in an infinite system of finite densities, the transition requires spontaneous symmetry breaking and therefore in a tilted systems the force on the immersed crytal would lead to a movement of the whole crystal. In this case the atoms are pinned with respect to each other and do not overlap, therefore there is no many-body coherence in the pinned state. The dynamical properties of this system is the focus of our ongoing work, with the mean-field lattice possessing unique characteristics compared to the static lattice potential.

4 - When discussing some examples of first-order quantum phase transitions, the Authors may consider citing the following paper "Pathway toward the formation of supermixed states in ultracold boson mixtures loaded in ring lattices, Phys. Rev. A 100, 013609 (2019)" where the phase separation of an imbalanced quasi-1D Bose-Bose mixture is described to be either a first-order or a second-order quantum phase transition.

Response: We thank the referee for bringing this reference to our attention and we have included it in the introduction of the paper.

5 - Fig. 1, panel (a), insets: both the solid green and the solid blue lines seem to be discontinuous functions (jump-like discontinuity). If I understood correctly, the energy per particle should be a continuous function of $g$, while its first derivative is discontinuous. Please briefly explain this in the main text.

Response: The referee is correct in stating that the energy per particle is a continuous function of $g$ with its first-derivative showing a discontinuity as a result of the first-order phase transition. The jumps observed in panel (a) of Fig. (1) are a result of the finite system size instead and the insets in panel (b) of the same figure show the expected behavior since the larger interaction strength of $g_m=3$ leads to a stronger localization of the immersed component for which the finite box size is negligible. We have emphasized the explanation of these finite-size effects and the resulting discontinuity in Fig. (1) (a) in the main text.

6 - In Fig. 2, the Authors plot $\rho(x)$. It may be worth plotting also the corresponding background densities $|\psi(x)|^2$.

Response: The density-density interactions ensure that the background density of the BEC is the inverse of the immersed particles density as shown in Fig.2 of the manuscript. As this would mirror the immersed particle density we decided to omit it for brevity.

7 - Fig. 3: I would add the labels "1/2" and "1/3" to the two horizontal segments.

Response: We thank the referee for this suggestion and have changed Fig. 3 accordingly.

8 - Concerning the description of $\rho(x)$ as a function of $g$ (see, e.g. Fig. 2), and the associated coherence properties, the authors never mention the concept of Mott-insulating state. Is there a reason why the authors omit this term? I think that, if it is correct, it may help the reader to understand the phenomenology at stake in a very effective way.

Response: We thank the referee for pointing this out. The self-pinned state that we describe in this work is conceptually similar to the Mott-insulating state, indeed the phase diagram we present in Fig.4 is reminiscent of that from the experiment by Haller et al., Nature 466, 597 (2010), where in our work the finite interspecies interaction $g_m$ replaces the lattice depth. To make this connection more clear we have added a sentence to the introduction and when the pinned state is introduced in Section 3.

9 - Concerning the finite-size effects in the phase diagram, the Authors say " it is in fact a result of finite size effects stemming from the particles hitting the edge of the box potential". Would periodic boundary conditions improve the result? I understand that it may be computationally more difficult to implement, but, I think that a brief comment about the difference open/periodic boundary conditions would be useful.

Response: Since in the infinite limit described by periodic boundary conditions the pinned state is only pinned with respect to each particle's neighbour (See answer above), the appearance of the pinned state requires spontaneous symmetry breaking. This is difficult to simulate and we have therefore opted to rather break the symmetry by hand by introducing a box potential. As the referee says, this also significantly reduces the numerical complexity.

10 - Fig. 2: Can the authors comment on the spacing between the $N$ peaks visible at high values of $g$?

Response: In the self-pinned state at high values of $g$ the density peaks of the immersed component are equally spaced on the order of $L/N$. However, in our simulations their positions strongly depend on the initial densities used. If one considers for example an adiabatic ramp from $g_m = 0$ to some finite value $g_m>0$ at a large and fixed value of $g\rightarrow\infty$ in the Tonks-Girardeau regime, then these positions are approximately given by the maxima of their initial density in the infinite box potential, which are determined by the odd solutions of $(2N+1)\tan(\pi z) = \tan[(2N+1)\pi z]$ for $0 \leq z \leq 1$ as detailed in our previous publication [8].

---

## Round 2 · Referee Report · Andrea Richaud (Referee 2) · 2023-5-16

Strengths

1) Interesting new first-order quantum phase transition; 2) Motivation and relevance for ongoing research are stated very clearly; 3) Good combination of analytical and numerical techniques; 4) Extensive and careful analysis, encompassing several indicators; 5) Effective presentation of the results

Weaknesses

Analysis restricted to few immersed atoms ($N=2$ and $N=3$), even if larger values of $N$ would lead to a remarkable computational complexity;

Report

After reading the Authors' reply to my comments and the new version of the manuscript, I believe that the Authors have carefully addressed all the points which I raised and have therefore improved the readability of the manuscript. I particularly appreciated their further comments about finite-size effects (a topic that was touched upon also by the other Referee) and about the analogy with the Superfluid to Mott-insulator transition

At this stage, I recommend the manuscript for publication.

---

## Round 2 · Referee Report · Anonymous (Referee 1) · 2023-5-17

Report

The Authors have appropriately addressed many of the 28 questions I raised in my previous report. Still, I believe some questions that might be answered better.

The numbering of the comments is the same as in the previous report.

  1. The minority particles are confined to a hard-walled box. This typically induces a finite-size effect in the energy of the order of 1/N. Instead using periodic boundary conditions seems to be much more natural as (a) finite-size effects would scale much more favorably, as $1/N^2$ (b) the geometry will be much more similar to that which appears in the thermodynamic limit, N→∞. Is it possible to use periodic boundary conditions for the minority component?

Response: Since in the infinite limit described by periodic boundary conditions the pinned state is only pinned with respect to each particle's neighbour, the appearance of the pinned state requires spontaneous symmetry breaking. This is difficult to simulate and we have therefore opted to rather break the symmetry by hand by introducing a box potential. This also significantly reduces the numerical complexity of the problem and allows to more accurately pinpoint the transition from superfluid to pinned.

If the transition is similar to a phase separation, there should no be problems in using periodic boundary conditions. Note, that periodic boundary conditions do not describe the infinite limit, but rather a system on a ring, which still has finite-size effects. Potentially, the finite-size effects can be smaller in that case. The original question was if it is possible to use periodic boundary conditions for the minority component?

  1. As there are different possible ways to introduce dimensionless parameters in the considered problem (there are three coupling constants under consideration, also it is common to set $2m=1$ in 1D), it would be much clearer to have all reported quantities in the full units. It seems that the used unit of length is directly related to the $s$-wave scattering length. Maybe it should be used as the unit of length? Also, the phase diagram the way it is presented now seems to depend only on the coupling constant. I do not find the notation in which density is a number without units to be clear enough, (N/L = 1/4, etc). The same applies to other physical quantities.

  2. The Local Density Approximation refers to the assumption of a piece-wise constant potential As far as I understand, the LDA assumes that the chemical potential locally can be approximated as the sum of the external field and the chemical potential of a homogeneous system. There is no requirement for the assumption of a piece-wise constant potential. Please check this point.

  3. We use this particular notion in order to be consistent with our previous publication

The choice of notation used in the study is not clear and intuitive.

We have decided against using a notation like A and B or ↑ and ↓ commonly found in descriptions of two-component systems since these notations usually describe two equally sized components.

I do not see the point, having labels A and B implies nothing about the number of particles.

  1. Moreover, since we are considering a quasi-one-dimensional system the term condensate fraction might be misleading and also lead to confusion with regards to the background Bose-Einstein condensate.

This statement is contradicting itself, if the term "condensate" is already used in one dimension for the background BEC, why should it be misleading applied to the immersed component?

If the Authors still want to keep the "coherence" term, please add a note that it corresponds to the definition of the condensate fraction. BTW, the quantity looks more like a "coherence fraction".

  • validity: high
  • significance: high
  • originality: high
  • clarity: good
  • formatting: excellent
  • grammar: excellent

Author:  Tim Keller  on 2023-05-31  [id 3698]

(in reply to Report 2 on 2023-05-17)

The Authors have appropriately addressed many of the 28 questions I raised in my previous report. Still, I believe some questions that might be answered better. The numbering of the comments is the same as in the previous report.

Response: We thank the referee for their time in evaluating our response and updated manuscript, and are happy to see that they agree with many of the points in our reply. We hope that the following responses adequately answer the remaining questions.

1 - The minority particles are confined to a hard-walled box. This typically induces a finite-size effect in the energy of the order of $1/N$. Instead using periodic boundary conditions seems to be much more natural as (a) finite-size effects would scale much more favorably, as $1/N^2$ (b) the geometry will be much more similar to that which appears in the thermodynamic limit, $N\rightarrow\infty$. Is it possible to use periodic boundary conditions for the minority component?

Response: Since in the infinite limit described by periodic boundary conditions the pinned state is only pinned with respect to each particle's neighbour, the appearance of the pinned state requires spontaneous symmetry breaking. This is difficult to simulate and we have therefore opted to rather break the symmetry by hand by introducing a box potential. This also significantly reduces the numerical complexity of the problem and allows to more accurately pinpoint the transition from superfluid to pinned.

If the transition is similar to a phase separation, there should no be problems in using periodic boundary conditions. Note, that periodic boundary conditions do not describe the infinite limit, but rather a system on a ring, which still has finite-size effects. Potentially, the finite-size effects can be smaller in that case. The original question was if it is possible to use periodic boundary conditions for the minority component?

Response: While the referee is correct that periodic boundary conditions for continuous systems have some similarity with the thermodynamic limit, and while they also represent finite ring-geometries, for systems which take on a discrete structure this comes with a number of additional complications. Let us first reiterate that not only in free space, but also on a ring the particles are only pinned with respect to their respective neighbours, and for both situations the appearance of the pinned state requires spontaneous symmetry breaking. As we said before, doing this by introducing a finite size box potential is a very clean and clear way. Furthermore, experiments are inherently of finite size and box potentials are a common experimental tool these days. Secondly, introducing periodic boundary conditions forces a periodicity on the system that is related to the size of the numerical grid and requires an additional step of numerical optimisation of the grid size (i.e. find a grid size that minimises the energy of the system after the transition). Our numerical simulations are already at the boundary of what it possible, and we therefore cannot carry out the additional optimisation of the grid. Our solution to this numerical issue is a finite sized box and we clearly spell out the restrictions this brings with it. Additionally, we are not aware of any unwanted finite size effects appearing in our calculations that would be eliminated when going to a $1/N^2$ scaling. This quadratic nature of the enhanced scaling is much more important when the particle numbers are large, and for our cases of $N=2,3$ it does not play a particularly large role. Furthermore, with the exception of the gray shaded areas in the phase diagrams, generally the immersed component is localized strongly enough either in the superfluid or self-pinned phase that the nature of the boundary conditions does not play a role at all.
If the referee could point out to us any effects related to this we might have missed, we would be happy to discuss them and work on mitigating them. We hope that the above convinces the referee that simulations with periodic boundary conditions would need a very good reason to justify the significant numerical effort needed and also our work would not be as relevant to current experimental setups. The short answer to the referee's question is therefore that it is, of course, possible to use periodic boundary conditions for the minority comment, and we have added a sentence to the conclusions to point this out. For the current work however, we cannot see a reason that justifies the significantly increased numerical effort.

9 - As there are different possible ways to introduce dimensionless parameters in the considered problem (there are three coupling constants under consideration, also it is common to set $2m=1$ in 1D), it would be much clearer to have all reported quantities in the full units. It seems that the used unit of length is directly related to the s-wave scattering length. Maybe it should be used as the unit of length? Also, the phase diagram the way it is presented now seems to depend only on the coupling constant. I do not find the notation in which density is a number without units to be clear enough, ($N/L = 1/4$, etc). The same applies to other physical quantities.

Response: The referee is correct and there are various different ways to scale a system in order to make the underlying physics more clear. The effects we describe in our work mainly depend on the ratios of the different interaction strengths and not on their absolute values. Therefore, using full units or absolute values and fixing a reference scale a priori leads to an unnecessary loss of generality without providing a benefit in terms of clarity. While it is possible using any of the three coupling strengths as the reference scale for introducing dimensionless units, the background BEC intraspecies coupling $g_c$ is the natural choice since it is constant and set to $g_c=1$ throughout the manuscript, whereas both other coupling strengths $g_m$ and $g$ are varied. This rescaling and the units of length $x_0=\hbar^2/mg_c$ are commonly used in the literature [Martin et al., PRL 98, 020402 (2007) and PRA 77, 013620 (2008) or Helm et al., PRL 114, 134101 (2015)]. Furthermore, setting $2m=1$ would also require dimensionless units which we believe contradicts the point the referee is raising. As we have shown in the previous reply and added to the resubmitted version of the manuscript, choosing the reference scale from typical values for cold atom experiments leads to a one-to-one correspondence of the dimensionless lengths appearing in our manuscript and physical lengths in units of $\mu$m, i.e. the position axis shown e.g. in Fig. 2 c) depicts a range of $8$ $\mu$m in total and the dimensionless density of $N/L = 1/4$ used throughout the manuscript corresponds to a line density of $0.25$ $\left(\mu\mathrm{m}\right)^{-1}$ in that case. Giving lengths in units of the scattering length instead and using the same exemplary reference scale would mean plotting e.g. Fig. 2 c) on a position axis spanning $-800$ to $800$ scattering lengths which hardly seems like an improvement in terms of clarity. Therefore it is not a suitable length scale for the physics we discuss. Furthermore, we do not fully understand the referee's comment regarding the phase diagrams only depending on the coupling constants. Apart from the (equal) masses of both species and the intraspecies interaction of the condensate $g_c$, which are all considered constant throughout the manuscript, it seems natural that the system's state and therefore its phase diagram would be a function of the remaining two variables $g$ and $g_m$. We strongly believe that the units we have chosen are the most suitable and intuitive ones for our work, and we have provided context and details for their interpretation. We have furthermore edited the discussion on the units in the manuscript to provide more clarity on this point. Ultimately it is a personal choice which units one prefers, but we do not think that the referee's suggestion of full SI units would be helpful in understanding the underlying principles of the work we discuss.

10 - The Local Density Approximation refers to the assumption of a piece-wise constant potential As far as I understand, the LDA assumes that the chemical potential locally can be approximated as the sum of the external field and the chemical potential of a homogeneous system. There is no requirement for the assumption of a piece-wise constant potential. Please check this point.

Response: The wording piece-wise was chosen to indicate that locally the potential is constant, similarly to e.g. F. Riggio et al. in Phys. Rev. A 106, 053309 (2022). This is equivalent to the definition the referee states, found e.g. in G. E. Astrakharchik, Phys. Rev. A 72, 063620 (2005), i.e. locally having a homogeneous system.

13 - We use this particular notion in order to be consistent with our previous publication The choice of notation used in the study is not clear and intuitive.} We have decided against using a notation like A and B or $\uparrow$ and $\downarrow$ commonly found in descriptions of two-component systems since these notations usually describe two equally sized components. I do not see the point, having labels A and B implies nothing about the number of particles.

Response: As before, choosing labels is partly a personal preference. We have decided to use labels that give information about the different components and have made sure that these are clear and intuitive. One can make an argument that A and B are labels from a set of equal letters and therefore they are not the best suited for describing a system of non-equals. But this is hair splitting and not helpful. The important part is that the labels appropriately describe the respective quantities, are readable and are used consistently. All of these criteria are fulfilled by our choice. Throughout the manuscript we also regularly remind the reader of the labelling, referring to "interpecies interaction $g_m$" and "intraspecies interaction $g$".

21 - Moreover, since we are considering a quasi-one-dimensional system the term condensate fraction might be misleading and also lead to confusion with regards to the background Bose-Einstein condensate. This statement is contradicting itself, if the term "condensate" is already used in one dimension for the background BEC, why should it be misleading applied to the immersed component? If the Authors still want to keep the "coherence" term, please add a note that it corresponds to the definition of the condensate fraction. BTW, the quantity looks more like a "coherence fraction".

Response: Our manuscript shows how a quasi-one-dimensional quantum system immersed into a background BEC fermionizes as its intraspecies repulsion $g$ is increased. As we only consider a small number of immersed particles ($N=2,3$), it would be misleading to suggest that this few-body system ever forms a Bose-Einstein condensate. We are therefore reluctant to refer to the occupation of the lowest natural orbital as the 'condensate fraction', even though it is formally equivalent. To remain consistent with other works on strongly correlated few-body systems [see Sowiński and García-March, Rep. Prog. Phys. 82 104401 (2019) for a recent review] we elect to call this the coherence, but have also added a comment in the manuscript on its correspondence to the condensate fraction as the referee recommends. Furthermore, there is no requirement for the background BEC to also be one-dimensional and as the referee correctly states, its existence already implies that the condensate is quasi-one-dimensional at most. Therefore the strongly-correlated gas could also be immersed into a fully three-dimensional background BEC instead, but we do not treat this case.

Anonymous on 2023-06-08  [id 3718]

(in reply to Tim Keller on 2023-05-31 [id 3698])

To be constructive, I will limit myself to one comment and one request.

comment:

Additionally, we are not aware of any unwanted finite size effects appearing in our calculations that would be eliminated when going to a 1/N2 scaling. This quadratic nature of the enhanced scaling is much more important when the particle numbers are large, and for our cases of N=2,3 it does not play a particularly large role.

My point is that the energy of the one-dimensional ideal Fermi gas converges as 1/N in a hard-wall box, and 1/N^2 on a ring. For N=3 particles, the energy difference with the thermodynamic value is around 30% and 10%, correspondingly. In that sense using a system on a ring is preferable, especially for the small number of particles.

request: Regarding the dimensionless units, I kindly request that the reported quantities be presented in dimensionless combinations that incorporate the relevant dimensionful quantities. For example, if "g" represents a coupling constant, it should possess units of energy multiplied by length. Similarly, if "mu" denotes a chemical potential, it should have units of energy. Hence, I would appreciate the use of dimensionless combinations in the figures. Please also ensure the consistency of units in equations and text.

---

## Round 2 · Author Response

We thank the editor and the referees for their time and handling of our manuscript, which we would like to resubmit for consideration to SciPost Physics.
We also thank the referees for their positive feedback about our work's suitability for publication and for their helpful comments in their reports.
After carefully studying the reports we have revised the manuscript to address the suggestions of the referees.
In light of the changes performed to the body of the manuscript and the support received by the reviewers, we hope the current version of our manuscript is suitable for publication.

Sincerely,
Tim Keller (on behalf of the authors)

---

## Round 2 · List of Changes

We have modified the manuscript as stated in the replies to the individual points of the referee reports.
The following references have also been added to the manuscript:

[8] M. A. Cazalilla, Bosonizing one-dimensional cold atomic gases, J. Phys. B 37(7), S1 (2004), doi:10.1088/0953-4075/37/7/051.
[9] S. Coleman, Quantum sine-gordon equation as the massive Thirring model, Phys. Rev. D 11, 2088 (1975), doi:10.1103/PhysRevD.11.2088.
[18] A. Richaud and V. Penna, Pathway toward the formation of supermixed states in ultracold boson mixture loaded in ring lattices, Phys. Rev. A 100, 013609 (2019), doi:10.1103/PhysRevA.100.013609.
[41] T. Kinoshita, T. Wenger and D. S. Weiss, Observation of a One-Dimensional Tonks-Girardeau Gas, Science 305(5687), 1125 (2004), doi:10.1126/science.1100700
[56] L. Cao, V. Bolsinger, S. I. Mistakidis, G. M. Koutentakis, S. Kronke, J. M. Schurer and P. Schmelcher, A unified ab initio approach to the correlated quantum dynamics of ultracold fermionic and bosonic mixtures, J. Chem. Phys. 147(4) 044106 (2017), doi:10.1063/1.4993512

---

## Round 3 · Author Response

We thank the editor and the referees for their time in evaluating our response and updated manuscript and are happy to see that they agree with many of the points in our reply.
As stated in the latest reply to the referee report, we have again revised the manuscript to address the remaining points raised by the referee.
We hope that these changes together with our responses to the report adequately answer the remaining questions.

Sincerely,
Tim Keller (on behalf of the authors)

---

## Round 3 · List of Changes

- rephrased the introduction of the units in the model section
- added comment about the condensate fraction to the definition of the coherence
- added sentence about the possibility of using periodic boundary conditions to the conclusion

---

## Round 4 · Author Response

We thank the editor and the referees for their time in evaluating our response and updated manuscript.
We agree with the referee's statement about the finite size effects for the periodic boundary conditions.
Following the referee's request we have added units to the figures in terms of the physical combinations making up the dimensionless unit of each reported quantity.

Furthermore, in order to ensure consistency of units between the new figures and equations, we have also reintroduced physical units to all equations and quantities in the text, starting from the model eqs. (1).
This is also fulfilling the request raised in point Nr. 8 in the referee's original report.
We hope that these changes sufficiently answer the report.

Sincerely,
Tim Keller (on behalf of the authors)

---

## Round 4 · List of Changes

• added units to the figures
  • reintroduced physical units to the text

---

## Editorial Decision

published